# The Role of Bedload Transport in the Development of a Proglacial River Alluvial Fan (Case Study: Scott River, Southwest Svalbard)

Waldemar Kociuba 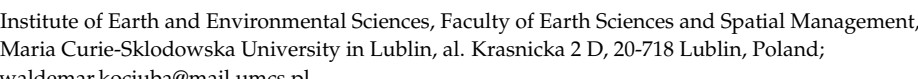

Institute of Earth and Environmental Sciences, Faculty of Earth Sciences and Spatial Management, Maria Curie-Sklodowska University in Lublin, al. Krasnicka 2 D, 20-718 Lublin, Poland; waldemar.kociuba@mail.umcs.pl

**Abstract:** This study, which was conducted between 2010 and 2013, presents the results of direct, continuous measurements of the bedload transport rate at the mouth section of the Scott River catchment (NW part of Wedel-Jarlsberg Land, Svalbard). In four consecutive melt seasons, the bedload flux was analyzed at two cross-sections located in the lower reaches of the gravel-bed proglacial river. The transported bedload was measured using two sets of River Bedload Traps (RBTs). Over the course of 130 simultaneous measurement days, a total of 930 bedload samples were collected. During this period, the river discharged about 1.32 t of bedload through cross-section I (*XS I*), located at the foot of the alluvial fan, and 0.99 t through cross-section II (*XS II*), located at the river mouth running into the fjord. A comparison of the bedload flux showed a distinctive disproportion between cross-sections. Specifically, the average daily bedload flux $Q_B$ was 130 kg day$^{-1}$ (*XS I*) and 81 kg day$^{-1}$ (*XS II*) at the individual cross-profiles. The lower bedload fluxes that were recorded at specified periods in *XS II*, which closed the catchment at the river mouth from the alluvial cone, indicated an active role of aggradation processes. Approximately 40% of all transported bedload was stored at the alluvial fan, mostly in the active channel zone. However, comparative Geomorphic Change Detection (GCD) analyses of the alluvial fan, which were performed over the period between August 2010 and August 2013, indicated a general lowering of the surface (erosion). It can be assumed that the melt season's average flows in the active channel zone led to a greater deposition of bedload particles than what was discharged with high intensity during floods (especially the bankfull stage, effectively reshaping the whole surface of the alluvial fan). This study documents that the intensity of bedload flux was determined by the frequency of floods. Notably, the highest daily rates recorded in successive seasons accounted for 12–30% of the total bedload flux. Lastly, the multi-seasonal analysis showed a high spatio-temporal variability of the bedload transport rates, which resulted in changes not only in the channel but also on the entire surface of the alluvial fan morphology during floods.

**Keywords:** bedload sampling; bedload flux; river bedload trap; proglacial gravel-bed river; sediment budgeting; Svalbard



## 1. Introduction

The High Arctic rivers that are devoid of human influence are sensitive indicators of contemporary environmental changes driven by climate change [1]. The effectiveness of these changes can be assessed by examining the component relationships of river transport [2–5]. In the polar regions, the total sediment yields from glaciated (600~40,000 t y$^{-1}$) and non-glaciated (500–1000 + t y$^{-1}$) catchments show great variability [6]. It is mainly determined by different ablation rates as well as the frequency of precipitation-driven high flows and floods [7], as well as outburst floods [8]. In particular, High Arctic proglacial rivers are dominated by sediment transport, where the contribution of bedload varies from <1% to 37% and is usually several times lower than the suspended load (from 2% to

81%) [6,9]. Bedload flux is characterized by greater temporal and spatial variability than suspended flux [6], in addition to the local character of sediment delivery to the channel (e.g., bank slumps) [10] and its dominance (70–90%) in the total sediment loads that are transported during flood events [6,11]. Bedload and its relation to other fluvial transport components (suspended and dissolved loads) is an important indicator for assessing the stage of a river system's development. The amount of bedload transported by a river also determines its channel geometry and reshaping ability under natural disturbances (e.g., caused by floods). The relation between flow activities and bedload transport rates has been highlighted by Arnborg et al. [12] and Williams [13]. Both field and laboratory experiments indicated that discharge and bedload transport relationships can be analyzed on the basis of hysteresis loops [14]. They found that for long-existing and continuous sediment concentrations associated with a long flood, the relationship between bedload transport and discharge in the relationship diagram is represented by a figure eight with a clockwise loop for high flows and a counterclockwise loop for low flows. Bedload transport measurements conducted during two spring freshets at Harris Creek (British Columbia, Canada) revealed that the bedload flux remained in a 'partial transport' regime and seasonal hysteresis. The results varied by study reach location and year [15]. The amount of discharged bedload, identifying its sources of supply and routes of distribution, as well as observing the conditions of its transport and deposition are all essential for carrying out a reliable assessment of contemporary trends that prevail in the development of a channel system [16].

There are difficulties with the methodology of making bedload measurements [17,18], and that translates to a low number of direct studies. Nevertheless, there are relevant studies from regions with cold-climate environments that emphasize the morphogenic role of bedload in the contemporary development of valley bottoms [19]. Moreover, there is research on the contemporary development of high-latitude river valleys that has focused on conducting storage assessment [20,21], noting sources of supply [9] and sediment budgeting [5] while also assessing the rates and directions of bed as well as channel form reshaping, based on high-resolution elevation models [22–25]. Furthermore, direct studies of bedload flux in the polar area are less frequently published in the respective scientific literature, although some examples can be found for both the High Arctic [26] and Antarctic regions [27,28]. These works are mainly concerned with the bedload transport mechanism [29,30] and the quantitative assessment of bedload flux under varying hydrodynamic stream conditions [31]. In light of this, it must be reiterated that one of the key reasons for having so few published studies in this field is the number of aforementioned difficulties associated with the measurement technique [32]. In practice, there is a considerable number of methods used to measure bedload transport, but there is only a small number of measurement sites and short measurement series (non-representativeness), all of which cause the published results described in the research literature to be hardly comparable [17]. Therefore, the effectiveness of measurements largely depends on the measurement device efficiency, particularly for the flow disturbances caused by the sampler [32], and at the same time, the comparability of results is ensured by the use of uniform devices and a standardized methodology [26].

The aim of this study is to determine the role that a mouth alluvial fan plays in bedload distribution. The locations of the measurement profiles above and below these depositional landforms make it possible to assess the spread of bedload particles as well as the potential of the fan to accumulate and redistribute bottom sediments. The novelty of this work is in reporting these data for use in studies of geomorphic changes, which are carried out in this area, based on differential analysis of high-resolution digital elevation models.

## 2. Study Sites and Methods

### 2.1. Study Area

The research into the Scott River alluvial fan reshaping was performed in the lower section of the small valley glacier catchment area (Figure 1A). The Scott River is located

in the northwest part of Wedel-Jarlsberg Land (southwest Svalbard), adjacent to Bellsund Bay and Recherche Fjord. The river catchment is contained within a 10.1 square kilometer (km$^2$) area that is 40% covered by a ~3 km long valley glacier (Figure 1A). The highest parts of the glacier reach ~600 m above sea level (a.s.l.), while the glacier snout reaches up to ~92.5 m a.s.l. The 3.3 km long unglaciated part of the valley [21] drains a river system that is fed by small tributaries and characterized by variable channel patterns [19] (Figure 1B). In the Svalbard area, the Scott River is representative of a typical gravel-bed river with a glacial regime in that the dominant (90%) feed source is water from the melting glacier [33]. The alluvial fan surface at the mouth of the gorge section of the Scott River valley covers an area of 0.06 km$^2$. The altitude differences range from 0.5 m to 5.8 m a.s.l. The length of the fan from the foot to the mouth is 300 m, and its maximum width is 450 m. The zone of an active channel occupies the northern part of the alluvial fan, forming a small lagoon from which the water is channeled along the shore rampart to an outlet located in the south-eastern part of the fan. The central and southern parts of the alluvial fan's surface are covered by a network of distribution channels, which fill up with water only during the biggest floods (Figure 1B).

### 2.2. Meteorological and Hydrological Monitoring

The meteorological station was located on a relatively flat elevated marine terrace at an altitude of about 23 m a.s.l. and 200 m from the Recherche Fiord shoreline, in close proximity to the study catchment and research station of Maria Skłodowska-Curie University (MCSU) (Figure 1A,B). This has been the permanent location of the meteorological station since the beginning of MCSU measurements in 1986. Meteorological measurements were conducted by an automatic meteorological station with a 10-min time span. The Scott River's discharge was measured at two water gauges located in the lower section of the catchment (Figure 2A). The first hydrometric profile, whose location followed previous hydrographic studies conducted by a team from MCSU [33,34], was located approximately 350 m above the river mouth to the Recherche Fjord. The hydrometric profile was located in a gorge crossing elevated marine terraces, where the valley floor narrows to about 50 m and the Scott River concentrates the braided channels into a single one. Hence, almost all surface waters from the Scott River catchment met the water gauge. The second was located in a single channel collecting water from the alluvial fan about 100 m from the river mouth to the Recherche Fjord. Pressure water level loggers were installed in the Scott River channel, recording changes in the water level and temperature within a 10-min time span. The river discharge was determined using a rating curve equation developed from periodic measurements of the flow velocity. The river flow was measured twice a week using a Hega II-type current meter and, in 2012 and 2013, an additional Acoustic Digital Current Meter (OTT ADC). No meteorological or hydrological data were available for the 2011 measurement season.

### 2.3. Bedload Sampling

Monitoring of the bedload transport was carried out by using two multi-module river bedload trap kits (Figure 2A). Such direct samplers, which are based on a bottom-adjacent frame and a mesh container that intercepts sediment particles moving along the bottom, enabled efficient measurements of the bedload transport rates to be made. Devices of this type are used by many countries' hydrological and geological services [17,18].

Regarding the hydrometric profiles, the bedload transport measurements were conducted in two cross-sections located in the lower section of the Scott River valley (Figure 2B). The first one (*XS I*) was located above the foot of the alluvial fan at about 2.5 km below the glacier terminus. The cross-section was located in the narrowing of the gorge section of the valley that cuts through elevated marine terraces (Figure 2B). This location was favored due to the fact that at the average and low water stages, the Scott River goes into a single channel. The second cross-section (*XS II*) was located at the wide single channel that collects water flowing from the alluvial fan, which is about 150 m away from the river mouth of the fjord (Figure 2A). The use of multi-module RBT sets enabled not only simultaneous but also

continuous multi-zone measurement with a flexible (up to 24 h), flow-adjusted sampling time. This was made possible by the large volume of the container and a maximum sample mass of up to 45–50 kg (optimum sample volume of up to 40% of the container capacity was about 18–20 kg). Correspondingly, both the effectiveness and the representativeness of the measurement were kept high by the fixed-distance locations of the measuring sites, which were evenly arranged every 1.5–2 m in the cross-section, and 8-point stabilization of the sampler. It must be noticed that the fast and easy (2–5 min) replacement of the RBT modules provided the means to maintain continuity of measurements in the adopted time cycle [35]. In all the analyzed melt seasons (2010–2013), bedload transport monitoring was conducted as continuous measurements at daily intervals (24 h).

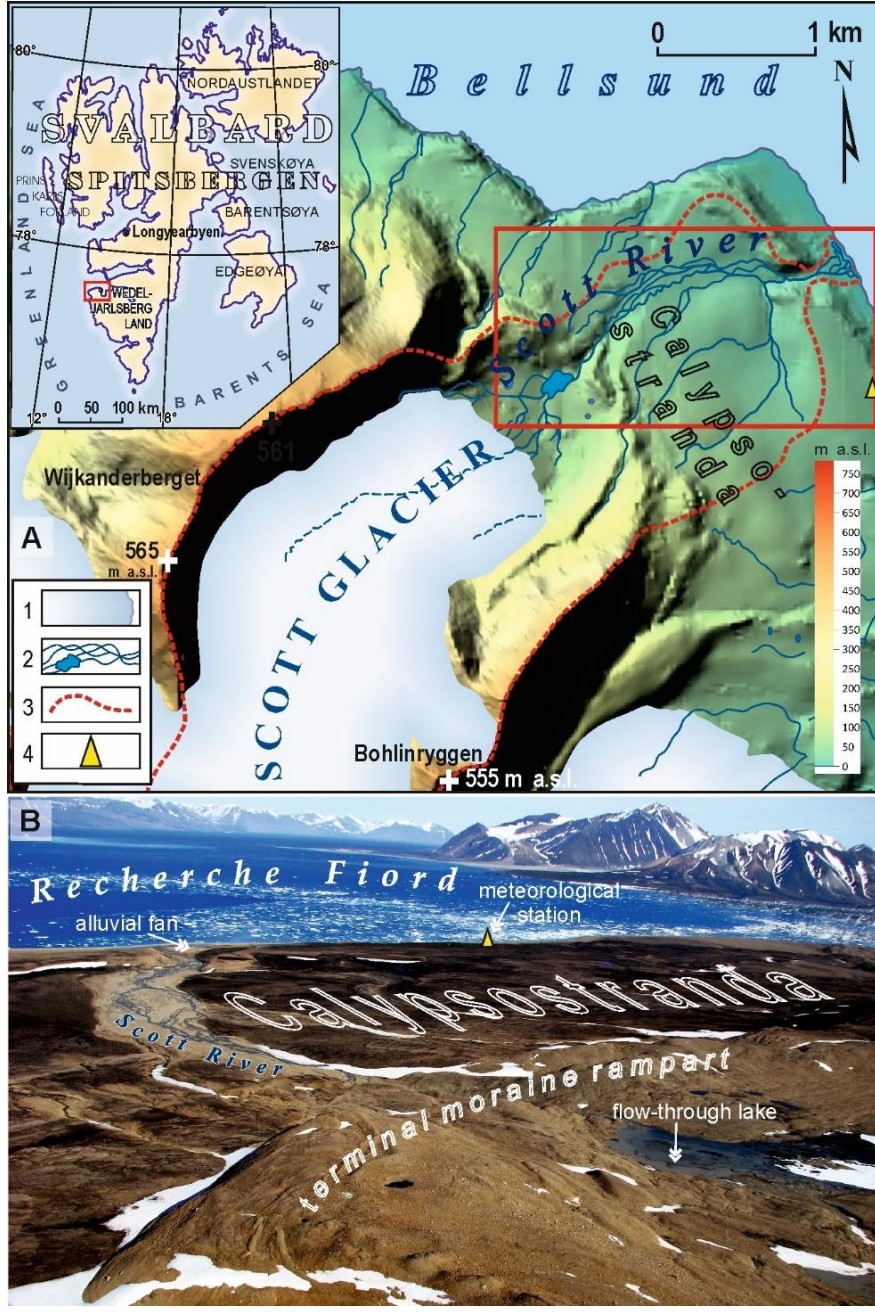

**Figure 1.** (**A**) Location of the study area in Svalbard and northwest Wedel Jarlsberg Land. 1: glaciers; 2: rivers and lakes; 3: catchment border; 4: meteorological station. (**B**) Downstream view of the Scott River valley.

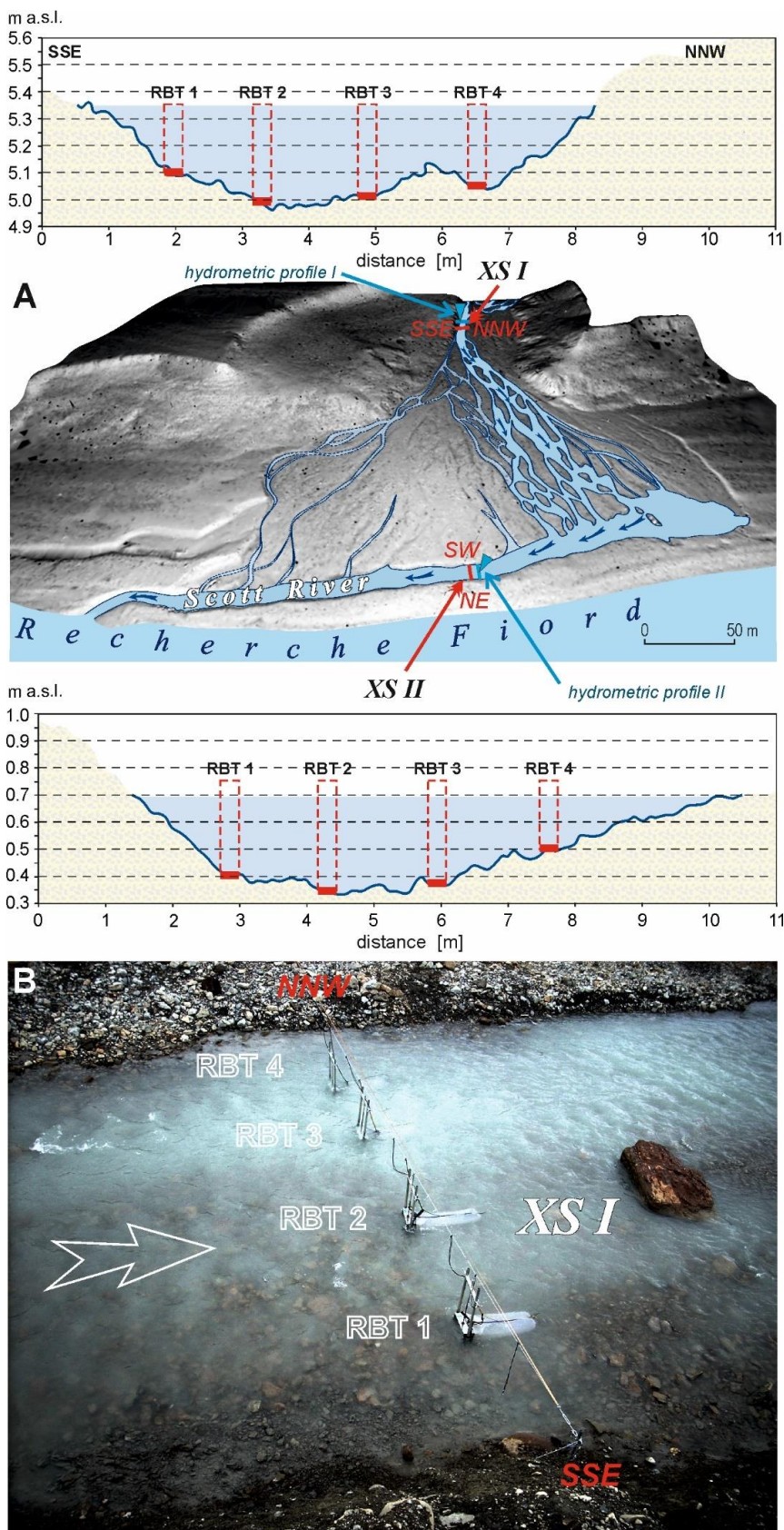

**Figure 2.** (**A**) Upstream view of the Scott River alluvial fan, with a sketch of cross-section I (*XS I*) and cross-section II (*XS II*) and the locations of the RBT modules. (**B**) Example of cross-section I (*XS I*) gauging.

In each cross-section, the bedload flux was investigated at a minimum of four evenly spaced measurement sites, which allowed for recognizing both the rate and the spatial variability of bedload transport across the monitored cross-section. The measured material of each cross-section was dewatered and weighed, both of which were performed separately for each measurement site.

Over the course of four consecutive melt seasons between 2010 and 2013, measurements were conducted at the same locations in the cross sections. The duration of the measurements in the individual seasons and cross-sections was varied, and it ranged from 24 to 59 days (Table 1).

**Table 1.** Basic parameters of measurements at Scott River cross-sections *XS I* and *XS II* during melt seasons in 2010–2013.

| Year | 2010 | | 2011 | | 2012 | | 2013 | | Total | |
|---|---|---|---|---|---|---|---|---|---|---|
| | *XS I* | *XS II* | *XS I* | *XS II* | *XS I* | *XS II* | *XS I* | *XS II* | *XS I* | *XS II* |
| Measurement period | Jul. 13 Aug. 10 | Jun. 27 Aug. 10 | Jul. 06 Aug. 29 | Jun. 21 Jul. 29 | Jul. 13 Aug. 24 | Jul. 13 Aug. 24 | Jul. 11 Aug. 13 | Jul. 11 Aug. 13 | - | - |
| Number of RBT modules | 4 | 4 | 4 | 5 | 4 | 4 | 4 | 4 | - | - |
| Measurement days | 29 | 45 | 24 | 39 | 43 | 43 | 34 | 34 | 189 | 161 |
| Samples collected | 114 | 177 | 96 | 193 | 142 | 126 | 109 | 110 | 618 | 606 |
| Sample size (kg) | 267 | 526 | 594 | 677 | 282 | 224 | 180 | 74 | 2080 | 1501 |
| Parallel measurement days | 29 | | 24 | | 43 | | 34 | | 130 | |
| Samples collected | 114 | 114 | 96 | 119 | 142 | 126 | 109 | 110 | 461 | 469 |
| Sample size (kg) | 267 | 252 | 594 | 438 | 282 | 224 | 180 | 74 | 1323 | 988 |

The reason for this was the different weather conditions affecting the length and timing of successive snowmelt seasons. The feasibility of installation and stability of operation of the RBT modules depended on the rate of permafrost withdrawal from the stream bed. In addition, the active layer's thickness needed to range from 0.2 m to optimally 0.3–0.4 m for the operation to be possible and effective. The period of complete snow cover melting, which sometimes persists in this part of the bottom of the valley even until the first half of July, was also an important factor behind the commencement of *XS I* measurements (Figure 3A). Despite the short distance between cross-sections, snow cover melting in the vicinity of *XS I* occurs much later due to shading of the valley by the gorge section's steep slopes (Figure 3B).

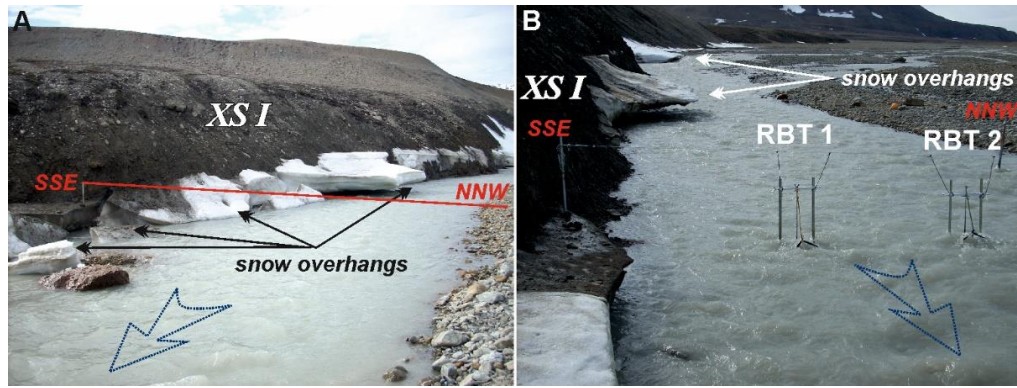

**Figure 3.** Upstream view of the Scott River cross-section I (*XS I*). (**A**) Snow overhangs visible on the right bank of the Scott River (north aspect of the slope), impeding the initiation of bedload transport monitoring. (**B**) Remnants of snow overhangs in cross-section I (*XS I*), visible but partly covered by unconsolidated sediment deposited on the snow surface.

On account of this, the compilation compared only periods when measurements were being carried out simultaneously in both cross-sections. These measurements were made in line with a daily cycle (every 24 h) of fixed time points (from 10:00 a.m. to 12:00 p.m.

UTC). Next, the bedload transport rate $q_b$ (kg m$^{-1}$ d$^{-1}$) for all the measuring sites in both cross-sections was measured according to the following Formula (1):

$$\text{Bedload transport rate } (q_b) \; q_b = \frac{G_s}{S_w T} \; [\text{kg m}^{-1} \text{ d}^{-1}] \tag{1}$$

where $G_s$ is the sample weight (in kg), $S_w$ is the sampler width (in m), and $T$ is the total sampling time (in 24 h).

The cross-section bedload flux $Q_B$ was calculated as the product of the transport rate $q_b$ and cross-section width $w_s$ (in m).

## 3. Results

The 2010 and 2012 summer measurement periods were slightly cooler and drier, while 2013′s was warmer and wetter than the multi-year average from 1986 to 2009 [36]. During the 2010 summer season, daily air temperatures ranged from 1.9 °C to 6.2 °C; the average was 4.5 °C [26]. During the measurement period, only 7 days with precipitation were recorded. The total precipitation in the summer measurement period (8.6 mm) was slightly lower than the multi-year value [36]. The highest daily sum of 3.0 mm occurred on 20 July (Figure 4).

The mean daily air temperature in 2012 was similar to 2010 at 4.6 °C, while in 2013, it was much higher (5.9 °C). In the 2012 summer measurement period, daily air temperatures ranged from 2.0 to 7.1 °C, while in 2013, they were from 2.7 to 8.6 °C (already after the end of the measurement period by 17 August). The 2012 and 2013 measurement periods also differed in terms of precipitation. The total precipitation in the summer of 2012 was 26.7 mm, and in the summer of 2013, it was more than three times higher (98.9 mm) [31]. In the 2012 measurement period, precipitation occurred infrequently and was characterized by high variability. Fourteen days with precipitation were recorded. The highest daily precipitation sum of 11.3 mm was recorded on 22 July. This was almost a half of the total precipitation in the 2012 measurement period and exceeded the total precipitation of the 2010 measurement period. In the summer of 2013, precipitation was more abundant and occurred more often than in previous years. Twenty days with precipitation were recorded, and the highest daily sum of 16.8 mm occurred on 13 August (Figure 4). Meteorological conditions caused the course of hydrological processes in the 2012 and 2013 measurement periods to also vary and differ significantly from the summer 2010 period. During the 2010 measurement period, low variability in the water levels and flows in the Scott River channel was recorded. The mean daily discharges at *XS I* ranged from 0.8 to 1.7 m$^3$ s$^{-1}$, and those at *XS II* ranged from 0.6 m$^3$ s$^{-1}$ to 0.9 m$^3$ s$^{-1}$ (Figure 4). In both measurement profiles, the mean daily discharges were about 1.2 m$^3$ s$^{-1}$ and were higher than the long-term average by 0.3 m$^3$ s$^{-1}$ [29]. As a result of the three-day flood, which culminated on July 12 [22], in both measurement profiles, the maximum mean values of the flows in the measurement period occurred on 13 July, when the temperature rise (up to 6.2 °C) was accompanied by precipitation of 1.8 mm (Figure 4). The highest daily precipitation occurred 18–20 July (up to 3.0 mm). Nevertheless, the simultaneous drop in temperature meant that no increase in the mean flow velocities was observed. The analysis of daily discharges in the summer season of 2012 shows that at the beginning of the measurement period, there was a gradual decrease in discharges to 0.40 m$^3$ s$^{-1}$ (*XS I*) on 6 August. In the following days, an increase was observed, with the culmination of a discharge of 2.19 m$^3$ s$^{-1}$ (*XS I*) on 10 August. The lowest discharge of 0.29 m$^3$ s$^{-1}$ was recorded at the end of the measurement period on 23 August (Figure 4). A steady increase in discharges with several floods was observed during the 2013 measurement period. The maximum discharge of 3.1 m$^3$ s$^{-1}$ during the measurement period was recorded on 22 July (Figure 4), while after the end of the measurement period, another flood took place, during which the maximum discharge of 4.63 m$^3$ s$^{-1}$ (*XS I*) was recorded on 16 August [31]. The flood wave was preceded by the lowest recorded water level in the Scott River during the melt season, resulting in a decrease in discharge to 0.74 m$^3$ s$^{-1}$ (13 August; Figure 4).

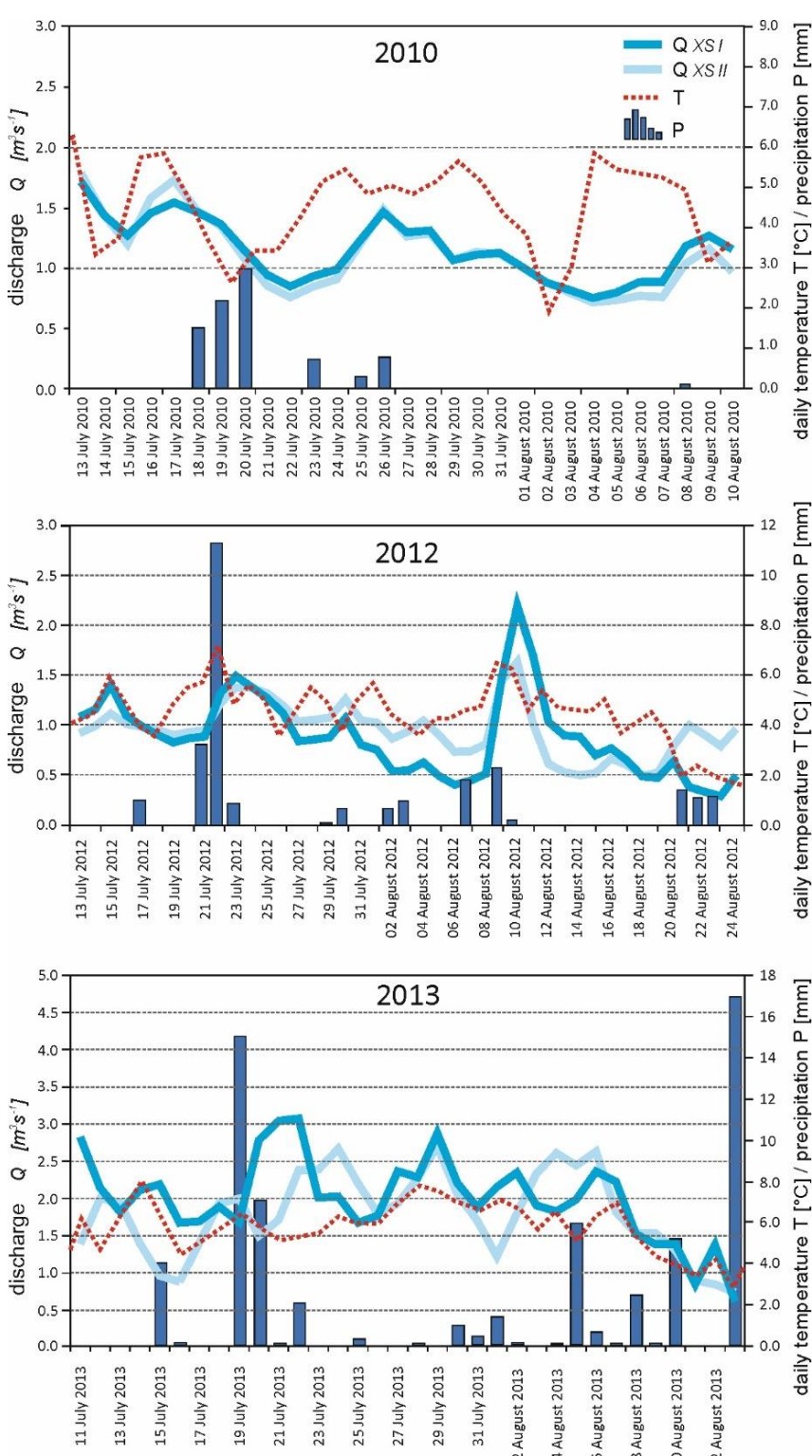

**Figure 4.** Hydrogram of daily discharge (*Q*) in the Scott River cross-sections *XS I* and *XS II* against the background of the mean daily temperature (T) and precipitation (P) in the measurement seasons of 2010, 2012, and 2013. No meteorological or hydrological data were available for the 2011 measurement season.

Spatio-temporal differences in discharge between the analyzed cross-sections resulted in different bedload flux during the periods analyzed. The comparison of daily bedload fluxes at *XS I* and *XS II* showed a very high variability both in the successive years and

during the measurement periods (Figure 5). Throughout the field study periods, the daily bedload flux $Q_B$ in the specified *XS I* and *XS II* cross-sections fell within the following ranges: from 0.1 (2012) to 1.5188 kg day$^{-1}$ (2011) at *XS I* and from 0.1 (2012) to 784 kg day$^{-1}$ (2011) at *XS II*. In addition, the recorded daily averages for *XS I* and *XS II* in successive years that spanned from 2010 to 2013 were as follows: 76.0 and 71.9; 242.3 and 126.6; 50.9 and 42.7; and 152.7 and 83.4 kg day$^{-1}$, respectively (Table 2).

**Table 2.** Daily bedload flux at Scott River cross-sections *XS I* and *XS II* during the melt seasons of 2010–2013.

| | 2010 | | | 2011 | | | 2012 | | | 2013 | | |
|---|---|---|---|---|---|---|---|---|---|---|---|---|
| | *XS I* | *XS II* | Difference | *XS I* | *XS II* | Difference | *XS I* | *XS II* | Difference | *XS I* | *XS II* | Difference |
| 6 July | | | | 292.7 | 98.4 | 194.3 | | | | | | |
| 7 July | | | | 100.1 | 69.8 | 30.3 | | | | | | |
| 8 July | | | | 66.0 | 49.0 | 17.0 | | | | | | |
| 9 July | | | | 37.0 | 10.6 | 26.4 | | | | | | |
| 10 July | | | | 1517.9 | 334.2 | 1183.7 | | | | | | |
| 11 July | | | | 1265.3 | 784.2 | 481.1 | | | | 39.8 | 0.3 | 39.5 |
| 12 July | | | | 504.9 | 570.2 | −65.3 | | | | 146.6 | 55.5 | 91.1 |
| 13 July | 658.2 | 499.3 | 158.9 | 85.4 | 329.2 | −243.8 | 1.1 | 7.4 | −6.4 | 82.6 | 42.1 | 40.4 |
| 14 July | 355.6 | 507.7 | −152.1 | 98.1 | 215.5 | −117.4 | 0.8 | 5.0 | −4.2 | 29.5 | 55.7 | −26.2 |
| 15 July | 107.3 | 171.1 | −63.8 | 160.0 | 139.2 | 20.8 | 2.6 | 6.8 | −4.2 | 95.5 | 53.9 | 41.6 |
| 16 July | 35.4 | 78.3 | −42.9 | 280.0 | 96.2 | 183.8 | 36.4 | 11.6 | 24.8 | 65.1 | 57.3 | 7.7 |
| 17 July | 338.6 | 320.5 | 18.1 | 13.3 | 10.7 | 2.6 | 1.1 | 1.0 | 0.1 | 5.7 | 9.1 | −3.4 |
| 18 July | 157.4 | 187 | −29.6 | 6.1 | 11.3 | −5.3 | 10.5 | 1.6 | 8.9 | 133.2 | 37.0 | 96.2 |
| 19 July | 34 | 79 | -45 | 11.0 | 58.3 | −47.3 | 3.2 | 0.4 | 2.8 | 104.8 | 5.1 | 99.7 |
| 20 July | 6.9 | 21.5 | −14.6 | 37.8 | 71.5 | −33.7 | 3.1 | 0.4 | 2.7 | 399.7 | 69.3 | 330.4 |
| 21 July | 2.2 | 10.5 | −8.3 | 678.2 | 106.6 | 571.6 | 2.8 | 1.1 | 1.8 | 648.5 | 34.0 | 614.5 |
| 22 July | 0.3 | 1.7 | −1.4 | 5.6 | 1.8 | 3.8 | 9.3 | 2.4 | 6.9 | 498.8 | 297.7 | 201.2 |
| 23 July | 0.3 | 1.2 | −0.9 | 2.4 | 8.7 | −6.2 | 374.0 | 49.5 | 324.5 | 644.9 | 334.9 | 310.0 |
| 24 July | 0.4 | 0.7 | −0.3 | 30.7 | 18.3 | 12.4 | 169.8 | 24.4 | 145.4 | 467.6 | 155.9 | 311.7 |
| 25 July | 0.5 | 0.8 | −0.3 | 9.9 | 9.9 | 0.0 | 282.6 | 41.4 | 241.3 | 443.6 | 309.0 | 134.7 |
| 26 July | 12.6 | 51.4 | −38.8 | 304.6 | 29.6 | 275.0 | 79.6 | 0.7 | 78.8 | 26.3 | 328.2 | −301.9 |
| 27 July | 57.3 | 95.7 | −38.4 | 290.4 | 6.4 | 284.0 | 11.4 | 2.9 | 8.5 | 56.6 | 244.8 | −188.2 |
| 28 July | 23.6 | 15.8 | 7.8 | 4.7 | 0.4 | 4.3 | 2.1 | 1.6 | 0.5 | 314.2 | 350.1 | −35.9 |
| 29 July | 24.1 | 6.4 | 17.7 | 12.1 | 7.2 | 4.9 | 3.4 | 1.4 | 2.0 | 262.7 | 192.5 | 70.2 |
| 30 July | 7.2 | 5.7 | 1.5 | | | | 134.1 | 41.3 | 92.8 | 126.8 | 66.7 | 60.1 |
| 31 July | 21.3 | 0.9 | 20.4 | | | | 18.7 | 14.9 | 3.8 | 73.2 | 12.8 | 60.4 |
| 1 August | 6 | 11.2 | −5.2 | | | | 4.1 | 10.6 | −6.5 | 72.4 | 9.7 | 62.7 |
| 2 August | 1.7 | 0.6 | 1.1 | | | | 2.0 | 5.1 | −3.1 | 38.1 | 5.9 | 32.2 |
| 3 August | 0.7 | 0.7 | 0 | | | | 1.0 | 0.3 | 0.8 | 127.4 | 5.2 | 122.3 |
| 4 August | 0.9 | 0.8 | 0.1 | | | | 0.7 | 0.3 | 0.4 | 106.2 | 24.7 | 81.5 |
| 5 August | 0.3 | 0.3 | 0 | | | | 0.4 | 0.1 | 0.2 | 82.2 | 6.7 | 75.4 |
| 6 August | 1 | 0.5 | 0.5 | | | | 0.6 | 0.3 | 0.3 | 0.3 | 0.4 | −0.1 |
| 7 August | 3.1 | 0.5 | 2.6 | | | | 0.4 | 0.2 | 0.2 | 80.4 | 48.6 | 31.8 |
| 8 August | 2.4 | 1.2 | 1.2 | | | | 0.4 | 0.2 | 0.2 | 0.4 | 17.3 | −16.9 |
| 9 August | 71.8 | 1.9 | 69.9 | | | | 27.2 | 1.2 | 26.0 | 1.4 | 3.0 | −1.7 |
| 10 August | 272.1 | 13.1 | 259 | | | | 271.9 | 270.9 | 1.0 | 2.0 | 0.5 | 1.4 |
| 11 August | | | | | | | 456.7 | 349.1 | 107.5 | 0.3 | 0.9 | −0.6 |
| 12 August | | | | | | | 91.1 | 249.8 | −158.7 | 0.5 | 0.2 | 0.3 |
| 13 August | | | | | | | 75.4 | 14.0 | 61.5 | 15.4 | 0.5 | 14.9 |
| 14 August | | | | | | | 3.1 | 33.3 | −30.2 | | | |
| 15 August | | | | | | | 0.6 | 62.3 | −61.6 | | | |
| 16 August | | | | | | | 1.5 | 162.0 | −160.5 | | | |
| 17 August | | | | | | | 1.2 | 108.8 | −107.6 | | | |
| 18 August | | | | | | | 0.6 | 64.6 | −64.0 | | | |
| 19 August | | | | | | | 0.9 | 174.7 | −173.7 | | | |
| 20 August | | | | | | | 0.8 | 49.8 | −48.9 | | | |
| 21 August | | | | | | | 0.5 | 60.0 | −59.6 | | | |
| 22 August | | | | | | | 0.1 | 1.3 | −1.2 | | | |
| 23 August | | | | | | | 0.3 | 1.4 | −1.2 | | | |
| 24 August | | | | | | | 98.9 | 0.7 | 98.2 | | | |
| Days | | 29 | | | 24 | | | 43 | | | 34 | |
| Sum | 2203 | 2086 | - | 5814 | 3037 | - | 2187 | 1837 | - | 5192 | 2836 | - |
| Min | 0.3 | 0.3 | −152 | 2.4 | 0.4 | −243.8 | 0.1 | 0.1 | −174 | 0.3 | 0.2 | −302 |
| Med. | 76 | 72 | 4 | 242 | 127 | 116 | 51 | 43 | 8 | 153 | 83 | 69 |
| Max | 658 | 508 | 259 | 1518 | 784 | 1184 | 457 | 349 | 324 | 649 | 350 | 615 |

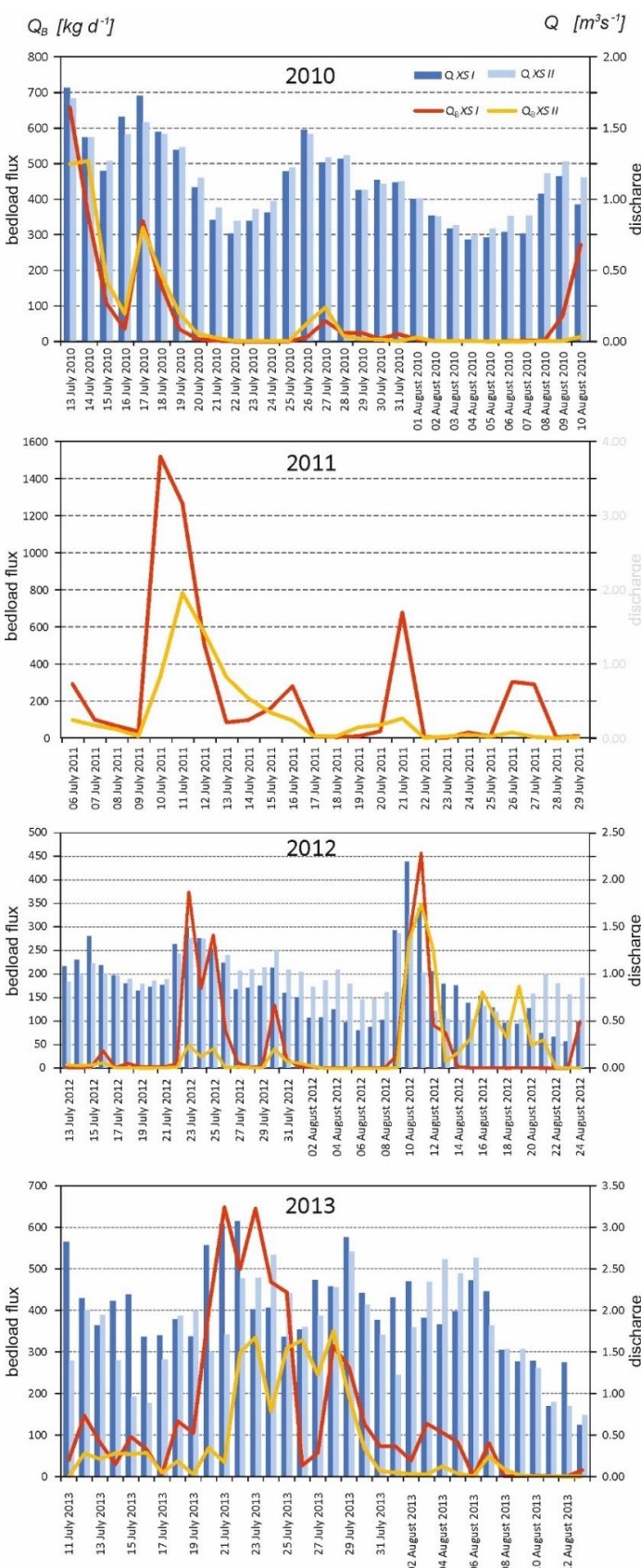

**Figure 5.** Changes of the bedload flux and water discharge at the Scott River cross-sections *XS I* and *XS II* in the measurement periods of 2010–2013. No meteorological or hydrological data were available for the 2011 measurement season.

Furthermore, the total bedload flux $Q_B$ that was discharged through the *XS I* and *XS II* cross-sections in each of the measurement periods was noted as follows: 2203 and 2086; 5814 and 3037; 2187 and 1837; and 5192 and 2836 kg, respectively. For each day, on average, bedload was carried away by *XS I* (130 kg) and *XS II* (81 kg) (Figure 5). Moreover, maximum daily loads of 658.2 and 507.7; 1517.9 and 784.2; 456.7 and 349.1; and 648.5 and 350.1 kg were recorded at both cross-sections over four consecutive melt seasons in the time of July's floods, and their exact dates were noted on 13 July, 10 July, 23 July, and 21 July, respectively (Table 2).

The share of maximum values in the total bedload flux varied from 12% (2013) to 30% (2010). Lastly, assuming that the bedload flux at *XS I* = 1, the decrease in the bedload flux at *XS II* was 0.9, 0.5, 0.8, and 0.5 in each of the subsequent melt seasons.

The spatial and temporal variability of the relation between bedload transport and water discharge for both cross-sections is presented in Figure 6. In all cases, periods of low and regularly rising flows are represented by a single loop, whereas higher and changing flows are represented by the irregular arrangement of loops forming figure eights. In 2010, the bedload flux was clearly related to the changes in discharge (Figure 4), and their mutual relationships are represented by counterclockwise loops. According to Arnborg et al. [12] and Williams [13], the relationships reflected by the figure eight with counterclockwise loops are characteristic of low flows. The greatest disparity in bedload transport rates between cross-sections was seen at the end of the measurement period (Figure 4), where an increase in discharge at *XS I* corresponded to the initial part of the reverse loop not occurring at *XS II* (Figure 6).

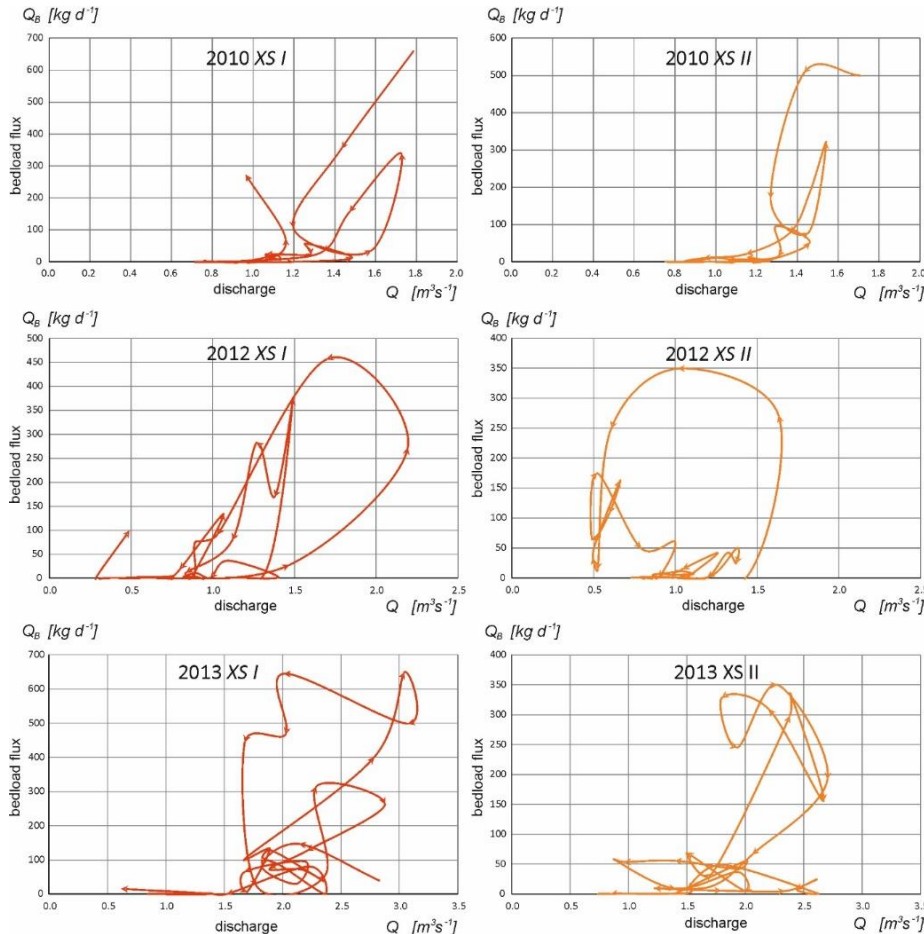

**Figure 6.** The bedload flux and water discharge relationships for the Scott River cross-sections in 2010, 2012, and 2013.

More complex relationships occurred during the 2012 measurement season. In the first decade, with fairly similar flows in both sections, a predominance of discharge in *XS I* was observed. In the second decade, in spite of the advantage of flows in the *XS II* section, the disparity was small and was observed mainly during the peak of flooding. Moreover, in the third decade, there was a decrease in flows with a predominance of discharge in *XS II* and an increase in flows in *XS II*, with a predominance of discharge in *XS I* (Figure 5). This fluctuation and spatial variation in discharge was reflected in the loop patterns (Figure 6). They were still inverse loops, but a single loop with a regular shape was marked at the beginning of the measurement period, and higher discharge values at *XS I* were responsible for flattening and shifting the loop into the higher flow range by 0.5 m$^3$ s$^{-1}$ (Figure 6). An even more different relationship between the bedload flux and discharge was found in the 2013 measurement season. Higher discharges in July 2013 than in 2010 and 2012 (Figure 4) were reflected in the occurrence of double clockwise loops, which Arnborg et al. [12] and Williams [13] indicated were characteristic of high flows. The 'concordant' loops fell during the two floods in July (20–24 and 27–29 July 2019). During the first of these, in addition, the largest discharge disparity between *XS I* and *XS II* occurred.

## 4. Discussion

The distribution of bedload depends on the climate-derived mechanism of fluvial transport [30]. Its identification is useful for recognizing the influence of flow conditions causing geomorphic changes in the channel [37]. The studies carried out in the Scott River catchment have documented a high variability of bedload flux both seasonally and over several years [38]. Although its share (3–8%) in relation to other components of fluvial transport is relatively low [6], bedload flux still plays a key role in the contemporary shaping of the valley bottom [39–44]. In the study period of successive years, there was a clear differentiation in the intensity of bedload transport that occurred during every melt season. Each year, there were identifiable periods of transport processes that were high in intensity, typically lasting until the middle or beginning of the second decade of July, and there were also periods of their intensity slowing down, which usually lasted from the second half of July to the beginning of September (Figure 5). In the first half of the melt season, the observations revealed a much higher bedload flux along with its high variability at the individual cross-sections. Then, the second half of the melt season showed a rather low daily bedload flux and lower variability between cross-sections (Figure 5). One of the main reasons for the seasonal variability in the intensity of bedload transport is the spatial and temporal variability of the sediment supply to the channel [21]. It must be noted that although the Scott Glacier provides as much as 90% of the water in the Scott River [33] and determines its flow conditions by releasing large amounts of sediments during the melt season's retreat, it provides little bedload material to the middle and lower valley sections. This is due to the concurrent interception of sediment by a functioning flow-through lake in the upper section of the valley floor [9,11,21]. Therefore, during the first half of the melt season, the channel in the lower section within the alluvial fan is directly fed with sediments that are delivered from the slopes and via snowbank overhangs in the gorge section of the valley floor through the elevated marine terraces. In the course of permafrost thawing and snow cover melting, mass processes are activated, and poorly consolidated sediments supply the river load [21]. These sediments are easily redeposited during snowmelt events and floods [32]. At the end of the melt season, as the valley slopes stabilize, the valley floor becomes the main source of sediment delivery, especially for the bottom and banks of the channels [10]. During this part of the melt season, this riverbed armoring [16] due to selective erosion of fine debris fractions [31] becomes a frequently observed phenomenon under the influence of decreasing flow velocities and extending the sediment transport distance. Riverbed armoring increases the velocity required to re-initiate the movement of bedload particles [30]. The clear overcharge of bedload at *XS I* relative to the amount recorded at *XS II* indicated the deposition of bedload particles within the alluvial fan (i.e., the predominance of aggradational processes during

the following measurement periods) [11]. However, the inter-seasonal analysis of bedload transport variability indicated that during periods of lowered flows, especially in the second half of the melt season, these proportions often reverse (Figure 4). This testifies to the stabilizing role of the alluvial fan, whose fleshy gravel cover intercepts a large part of the water resources as the melt season culminates [36]. The same instance is facilitated by the maximum thickness of the permafrost active layer that occurs at this time [21]. As a result, during the high-water stages, the alluvial fan becomes a reservoir of water [45,46], which is then released during the lows [31]. The consequence is the observed disproportion of bedload flux discharged by the analyzed cross-sections.

The results of many field and laboratory studies highlight how changes in the flow regime and hydrograph shape describe bedload transport and the resulting bed morphology [12–15]. Hassan and Church [15] highlighted that the hysteresis loops were more pronounced for hydrograms of longer durations, which corresponded to less instability until the peak discharge was reached. These observations were confirmed by the hysteresis of both cross-sections in the Scott River from 2012, with a clearly extended loop corresponding to a long period of slowly increasing flows and bedload transport intensity. Hassan and Church further noted that the hydrographs of shorter durations exhibited more time above the critical shear stress thresholds, leading to higher bedload transport rates and ultimately more variable hysteresis patterns [15]. The direction of bed surface adjustment was related to bedload hysteresis or, more specifically, clockwise hysteresis, which typically resulted in thickening of the bed. More frequent and shorter hydrographs resulted in greater relative changes in the channel in terms of slope, topographic variability, and surface texture [15]. This in turn was reflected in most of the timing of the Scott River flows, where we observed high variability and the occurrence of short episodes of surges reflected by small, irregular counterclockwise loops.

The results of the measurements, which were carried out in four successive melt seasons, indicated high temporal and spatial variability of the processes that shaped the alluvial fan surface. In the first half of each melt season, the high dynamics of the bedload flux and an alternating occurrence of maxima at *XS I* and *XS II* were accompanied by both increased deep erosion and processes of lateral as well as vertical growth of the sediments [21]. Then, during the second half of these melt seasons, there was an observable stabilization of channel forms that was marked by an absence of irregularities. Furthermore, seasonal lateral migrations of the transport paths in the channel cross-section profile are related to the prevailing displacement and time-varying conditions of sediment delivery to the channel [19]. As a result, bedload is alternately stored and then redeposited in the channel as well as on the alluvial fan surface. In fact, such fluctuations in bedload are characteristic of its transport mechanism, and the decisive factor for the amount of drained material is flooding, during which up to 70–90% of the bedload is carried away [6]. In the Scott River catchment, the greatest amount of bedload material was transported during ablative fall floods [36,47]. In the subsequent years, the proportion of maximum daily loads that were recorded during the culminating flood wave varied from 12 to 30% of the total mass of the elevated bedload material. In the first half of the melt season, 1–2 precipitation-ablation floods usually take place, with their culmination occurring around mid-July (Figure 4). In the course of these floods, 30–70% of the total bedload flux is discharged [38]. However, in the second half of the melt season, floods of the precipitation-ablation type are rare, but they can be very violent [36]. Instances of floods that were observed in the second half of August (2012 and 2013) showed that their morphological effects played a significant role in reshaping the alluvial fan [36,48]. Moreover, the floods coincided with the period of maximum thickness of the permafrost active layer, so both the slope and bottom material could be easily mobilized and feed the bedload [11,21].

In consideration of the specific disparity between the two analyzed reaches, it can be noticed that the Scott River fits well into the general model of alluvial fan development [49]. Due to the change in slope below the gorge section's narrowing [21], the river is free to spread and infiltrate the alluvial fan surface. Correspondingly, this reduces the flow

carrying capacity and becomes a direct cause for the discovered differences in bedload flux. However, having used differentiating high-resolution digital terrain models (DTMs) that were acquired from terrestrial laser scanning (TLS) as of July and August 2010 [48,50] and 2013 [21], it was found that there was dominance of erosion (82% of the surface) in terms of both area and sediment volume. The alluvial fan surface had been lowered by a mean of 0.03 m y$^{-1}$ [17]. In contrast, an opposite trend was discovered following a comparison of the DTMs as of July and August 2013, and the DTMs extracted from the TLS surveys performed on 18 August 2013 came after the subsiding of an extreme flood event that had occurred between 14 and 16 August 2013 [36]. The effects of the aforementioned flood event, which was the highest on record in this part of Wedel-Jarlsberg Land since 1993 [33], led to the conclusion that above-average hydro-meteorological events with a rapid course can dramatically change the trend of transformation from erosional to aggradational. The coupling of high rainfall and rising temperatures brought about a 4% increase in mould volume within the alluvial fan zone. Consequently, over this three-year time span of 2010–2013, as much as 1.5 times the average annual amount of sediment was eroded from the said area [21].

## 5. Conclusions

- The multi-seasonal analysis showed not only a clear disproportion in the amount of bedload flux ($Q_B$) that had been discharged through the cross-section located at the foot of the alluvial fan (*XS I*) and at the river mouth (*XS II*) but also its high temporal and spatial variability. Furthermore, the bedload flux $Q_B$ in individual cross-sections varied from close to zero to mostly several hundred kilograms per day, although there were also extreme instances exceeding one and a half thousand kilograms per day. The temporal variability of the bedload flux was characteristic for both studied cross-sections and indicated itself in the differentiation of the transport volumes recorded over consecutive years, as well as in the variable dynamics over the course of individual seasons.

- It was observed that the spatial differentiation of the bedload flux was evinced by a disproportion om the bedload flux in the cross-sections, which occurred despite there being the same or very similar flow conditions. In analogous measurement periods, smaller bedload fluxes were found in the measurement profile that closed the catchment at the outlet of the alluvial cone. This proves that there is a predominance of aggradational processes within this area. In turn, the inter-seasonal variations of this trend testify to a depletion of sediment resources and redeposition of bedload. The largest among such migrations of bedload material take place during floods and result in changes to the channel's geometry and its morphology, as well as the surface of the alluvial cone. Thus, it has been confirmed that floods play a determining role in bedload transport. The highest daily volumes recorded in subsequent seasons accounted for 12–30% of the total bedload flux.

- The disparity between the two reaches determined for the lower section of the Scott River valley floor was typical of conditions for alluvial fan development. However, precise three-year-long measurements of changes in the fan surface indicated that erosion predominated in this time span's overall sediment balance, and the lowered surface was periodically overlain (deposition predominance) during flash flood episodes of snowmelt and glacier melt (first half of the melt season), or it was of a thermal rainfall origin (second half of the melt season), which resulted in the differentiation of the alluvial fan surface as well as the differentiation of the bedload flux.

**Funding:** This research received no external funding.

**Acknowledgments:** This study was carried out in the Scott River catchment in the summer seasons of 2010–2013 with the participation of the University of Maria Curie-Skłodowska's Polar Expeditions Team. This study was supported by the scientific project of the National Science Centre 2011/01/B/ST10/06996 'Mechanisms of fluvial transport and sediment supply to channels of Arctic

rivers with various hydrological regimes (southwest Spitsbergen)'. The author would like to thank the expedition's participants for their help with all the work in the field and the reviewers for their valuable comments that helped to make the improvements that were essential to the successful completion of this work. The author is also grateful for the English language proofreading by Luke Boczkowski.

**Conflicts of Interest:** The author declares no conflict of interest.

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
