# Peer review of "The Role of Bedload Transport in the Development of a Proglacial River Alluvial Fan (Case Study: Scott River, Southwest Svalbard)"

_hydrology, doi:10.3390/hydrology8040173_

Round 1

Reviewer 1 Report

The Role of Bedload Transport in the Development of a Proglacial River Alluvial Fan (case study: Scott River, SW Svalbard)

Waldemar Kociuba

General comments:

The presented study focuses on a relevant topic in fluvial geomorphology. Fluvial bedload rates are still very problematic to measure and a number of challenges and problems still exist, which make it rather difficult to gather and include also quantitative data of fluvial bedload for instance in sediment budget studies.

Therefore the efforts made by the authors are much appreciated. However, the presented manuscript clearly lacks scientific depth and a more elaborated international scientific framework. A higher number of international and up-to-date references are missing. Many of the used references in the entire manuscript are from Polish colleagues, which is fine but it should be a more balanced ratio of references. Although the English language is good, moderate changes would certainly improve the manuscript. Please avoid formulations like “what is more” and “each of the said seasons”.  

The main problem with this manuscript is that it is too close to already published studies dealing with the same or a similar subject conducted at the same study site and therefore hardly offers any new information. Although the authors include a more comprehensive dataset covering four melt seasons, this is not enough to meet the requirements for novelty and originality of a manuscript.

In addition, the manuscript lacks essential information regarding detailed meteorological and hydrological conditions during the measuring periods. No information is given e.g. on air temperature, precipitation, discharge or flow velocity.    

The study site, the applied methods and also parts of the achieved results have been published in several papers before. The presented field data from 2010 to 2013 are no longer really up-to-date.

In conclusion, I can hardly see any new information in the manuscript. I’m also not convinced that the actual focus of the manuscript fits well into the scope of the intended Special Issue on “Observations on Water Resources”.

Detailed comments:

Line 14: delete the word “bedload” after samples

Keywords: gravel bid river, proglacial

Line 55: Not clear, please rephrase.

Line 97: correct square km

Line 118: Not clear, please rephrase.

Line 242: Not clear, please rephrase.

Table 1: Please check! Rows are shifted and squeezed!

Figure 3 is labeled as Fig 24? Figure description is unclear. Please check!

I clearly recommend to present the stated bedload flux values in a table. This will make things much more easier for the reader.

Figure 4: Would be good to have the dates of the mentioned floods included in this figure. The red arrow is a bit confusing.

Author Response

Response to the suggestions and comments of the Reviewer #1

General comments:

R#1: The presented study focuses on a relevant topic in fluvial geomorphology. Fluvial bedload rates are still very problematic to measure and a number of challenges and problems still exist, which make it rather difficult to gather and include also quantitative data of fluvial bedload for instance in sediment budget studies. Therefore the efforts made by the authors are much appreciated. However, the presented manuscript clearly lacks scientific depth and a more elaborated international scientific framework.

R#1: A higher number of international and up-to-date references are missing. Many of the used references in the entire manuscript are from Polish colleagues, which is fine but it should be a more balanced ratio of references.

A: I agree. References were verified and supplemented.

R#1: Although the English language is good, moderate changes would certainly improve the manuscript. Please avoid formulations like “what is more” and “each of the said seasons”.

A: English language content was corrected.

R#1: The main problem with this manuscript is that it is too close to already published studies dealing with the same or a similar subject conducted at the same study site and therefore hardly offers any new information. Although the authors include a more comprehensive dataset covering four melt seasons, this is not enough to meet the requirements for novelty and originality of a manuscript.

A: I have supplemented the text with data have not been published anywhere before (Table 2 with daily bedload discharges for all 4 seasons), new analyses (including water discharge and bedload flux relationships), and their interpretations.

R#1: In addition, the manuscript lacks essential information regarding detailed meteorological and hydrological conditions during the measuring periods. No information is given e.g. on air temperature, precipitation, discharge or flow velocity.

A: I have supplemented the text with meteorological and hydrological background. Unfortunately, lack of hydrometeorological data from 2011, caused that this period was not analysed.

R#1: The study site, the applied methods and also parts of the achieved results have been published in several papers before. The presented field data from 2010 to 2013 are no longer really up-to-date.

A: I generally agree with this opinion. This is not 'new data' and continuous  bedload flux measurements in the Scott River were not continued in later years. However, this method is used in current studies in Antarctica and NW Svalbard. I refer to the published results of the former in the text. Furthermore, although these are not new measurements data their interpretation may be subject to change as the results of new studies, including remote sensing, become available. Also new to this paper is the reference to the results of TLS measurements, which shed new light on changes in alluvial fan morphology.

R#1: In conclusion, I can hardly see any new information in the manuscript. I’m also not convinced that the actual focus of the manuscript fits well into the scope of the intended Special Issue on “Observations on Water Resources”.

A: New data and interpretations of the results have been added. I agree about the choice of SI, but at the time of submission, it was the topic closest to the range of the proposed paper. I suggested to the editors that if in doubt, the manuscript could be reclassified to another topic.

All comments and suggestions contained in the review have been taken into account.

Detailed comments:

Line 14: delete the word “bedload” after samples

A: deleted “bedload” after samples

Keywords: gravel bid river, proglacial

A: replaced by: proglacial gravelbed river

Line 55: Not clear, please rephrase.

A: deleted sentence

Line 97: correct square km

A: replaced by: 10.1 square km (km2)

Line 118: Not clear, please rephrase.

A: replaced by: Bedload sampling

Line 242: Not clear, please rephrase.

A: rephrased on: During this part of the melt season that river-bed armouring [16], due to selective erosion of fine debris fractions [31], becomes a frequently observed phenomenon under the influence of decreasing flow velocities and extending sediment transport distance. River-bed armouring increases the velocity required to re-initiate the movement of bedload particles [30].

Table 1: Please check! Rows are shifted and squeezed!

A: Table columns and rows are aligned

Figure 3 is labeled as Fig 24? Figure description is unclear. Please check!

A: Corrected no of Figure to 3. The caption rephrased on: Upstream view of the Scott River cross-section I (XS I); (A) snow overhangs visible on the right bank of the Scott River (north aspect of slope), impeding the initiation of the bedload transport monitoring; (B) cross-section I (XS I); remnants of snow overhangs visible with partly covered by unconsolidated sediments deposited on the snow surface.

I clearly recommend to present the stated bedload flux values in a table. This will make things much more easier for the reader.

A: The Table 2 has been added: Table 2. Daily bedload flux at Scott River cross-sections XS I and XS II during melt seasons 2010-2013.

Figure 4: Would be good to have the dates of the mentioned floods included in this figure. The red arrow is a bit confusing.

A: Added new figure 4: Figure 4. Hydrogram of daily discharge [Q] in the Scott River cross-sections XS I and XS II in against the background of mean daily temperature [T] and precipitation [P] in the measurement seasons 2010, 2012 and 2013. No meteorological and hydrological data are available for the 2011 measurement season. On figures 4-6 at the horizontal axes, numbers have been replaced by dates

Reviewer 2 Report

Abstract little long but very accurate with a short indications on the principle results.

Keywords : repetition of bedload

Introduction well written and give more détails on the topics and what it’s said by the scientific communauty on yhe subject

Map and illustration very accurate and well drawn

Methodology well developped but the authors haven’t give any information on the way used to choose the sections where the measurements was done

The results are wel presented but the graphics in 3D shows little difficulties to compare the profiles.

No word on the climatics parameters to see what happen during these 3 years despite that authors indicate the importance of that in line 184

All results could be synthetised in table just de have a global eye on them

Discussion is very wide and well developped indicating the importante changes between seasons and years and traduce clearly the difficulties to make an accurate approach

Bibliography very importante

Author Response

Response to the suggestions and comments of the Reviewer #2

Thank you for your expressed opinions and valuable comments. I tried to include them in the revised version as much as possible.

General changes to the text

  1. References were verified and supplemented
  2. All figures was reedited according to Reviewers’ comment
  3. The Table 2 has been added
  4. The text has been redrafted and supplemented where possible. The meteorological and hydrological description has been added.
  5. English language content was also corrected.

Reviewer 3 Report

The Role of Bedload Transport in the Development of a Pro-glacial River Alluvial Fan (case study: Scott River, SW Sval- bard)

The manuscript presents a fieldwork study carried out to determine the bedload transport rates of Scott River. Bedload field observations were used in this work. The research herein presented is certainly within the scope of Hydrology.

According to my observations, the topic of the manuscript is interesting and challenging. However, the lack of clarity in some parts of the text should be fixed before the publication. I think the paper requires sharpening in the definition of the results obtained and subsequent discussion. Nonetheless, I am supportive with the manuscript and after the revision herein purposed I think it should be ready for publication. I will be happy to review an updated version of the manuscript.

List of comments

- Line 43 is rather vague and too general. What do you mean by “other fluvial transport components”?. I would include here the relation between discharge and sediment fluxes (i.e. sediment hysteresis, see references [1,2]). Sediment hysteresis serves as proxy to determine whether the fluvial channel operates under supply-limited sediments conditions. This is ultimately related with the aggradation/degradation processes occurring in the channel reach (i.e. the river system development).

- Line 101: It should read alluvial fan surface.

- Figure 1A. Include the location of the monitoring stations.

- Line 109. There is a small typo in “…River Beload Trap”.

- The Methods section needs to be expanded. Several key points related to the monitoring campaign are not explained:

  • How were the sections chosen?. Based on which criteria?. Did the author assess the uniformity of the flow velocity?, how meaningful is the monitoring river cross section with respect to the whole river reach?, do the measuring section belong to a straight patch of the river?, did the author check the presence of upstream boulders or pools which could bias the results?.
  • Figure 2. Based on which criteria the RBT modules were spaced along the cross section?
  • For how long the RBT modules were placed in the river to measure bedload fluxes?
  • Which is the minimum and maximum sediment size collected by the RBT modules?.
  • Why 4 or 5 RBT modules (see Table 1) were chosen?

- Provide the correct format to formula [1].

- Figure 4. Remove the 3D effect from the plot.

- Figure 4. Include a x-axis with the months. The current measurement-day x-axis is hard to interpret.

- Figure 4 needs to be interpreted in terms of annual precipitation and annual snow cover. This is the only way to distinguish any anomaly driven by the climatic forces.

- Results. Nothing is said about the discharge and sediment sizes. Is there any information on this matter?

- Discussion. I think the author should explore the bedload hysteresis patterns associated to the unsteady flows [2,3]. Such way, each climate-derived mechanism of fluvial transport could be related to a specific type of sediment hysteresis.

Bibliography

[1] The impact of hydrograph variability and frequency on sediment transport dynamics in a gravel-bed flume. Plumb B.D., Juez C., Annable W.K., McKie C.W. & Franca M.J. EARTH SURFACE PROCESSES AND LANDFORMS. 2019. DOI: 10.1002/esp.4770.

[2] The origin of fine sediment determines the observations of suspended sediment fluxes under unsteady flow conditions. C Juez, MA Hassan, MJ Franca. WATER RESOURCES RESEARCH. 2018, 1-16.

[3] Sensitivity of bed load transport in Harris Creek: Seasonal and spatial variation over a cobble-gravel bar. Hassan, M. A., and M. Church. WATER RESOURCES RESEARCH, 37(3), 813-825, doi:10.1029/2000WR900346.

Author Response

Response to the suggestions and comments of the Reviewer #3

General comments:

R#2: According to my observations, the topic of the manuscript is interesting and challenging. However, the lack of clarity in some parts of the text should be fixed before the publication. I think the paper requires sharpening in the definition of the results obtained and subsequent discussion.

A: The text has been partially redrafted and supplemented where possible. The meteorological and hydrological description has been added.

R#2: Nonetheless, I am supportive with the manuscript and after the revision herein purposed I think it should be ready for publication. I will be happy to review an updated version of the manuscript.

A: Thank you for all your comments and suggestions.

All comments and suggestions contained in the review have been taken into account.

Detailed comments:

- Line 43 is rather vague and too general. What do you mean by “other fluvial transport components”?. I would include here the relation between discharge and sediment fluxes (i.e. sediment hysteresis, see references [1,2]).

A: I have supplemented and expanded this section of the text by referring to the two proposed papers 1 and 3 (in text 14 and 15) and also the two earlier ones on hysteresis (12, 13). The section now reads: ‘Bedload and its relation to other fluvial transport components (suspended and dissolved loads) is an important indicator for assessing the stage of a river system’s development. The amount of bedload transported by a river also determines its channel geometry and ability to reshapement under natural disturbances e.g. caused by floods. The relation between flow activities and bedload transport rates has been highlighted by Arnborg et al. [12] and Williams [13]. Both field and laboratory experiments indicate that discharge and bedload transport relationships can be analysed on the basis of hysteresis loops [14]. They found that for long existing and continuous sediment concentrations associated with a long flood, the relationship between bedload transport and discharge in the relationship diagram is represented by a figure-eight with a clockwise loop at high flows and a counter-clockwise loop for low flows. Bedload transport measurements conducted during two spring freshets at Harris Creek (British Columbia, Canada) revealed that bedload flux remains in a 'partial transport' regime and seasonal hysteresis. Results varied by study reach location and year [15].’

- Sediment hysteresis serves as proxy to determine whether the fluvial channel operates under supply-limited sediments conditions. This is ultimately related with the aggradation/degradation processes occurring in the channel reach (i.e. the river system development).

A: In the ‘results’ chapter, I added fig. 6 ‘Figure 6 The bedload flux and water discharge relationships for Scott River cross-sections in 2010, 2012 and 2013.’, and in ‘discussion’ a paragraph related to its interpretation. Unfortunately, lack of hydrometeorological data from 2011, caused that this period was not analysed.

- Line 101: It should read alluvial fan surface.

A: added ‘surface’

- Figure 1A. Include the location of the monitoring stations.

A: All figures have been redrawn and missing elements have been added: the location of the monitoring stations added on Fig. 1 A and 1 B

- Line 109. There is a small typo in “…River Beload Trap”.

A: corrected typo in ‘Bedload’

- The Methods section needs to be expanded. Several key points related to the monitoring campaign are not explained: How were the sections chosen?. Based on which criteria?. Did the author assess the uniformity of the flow velocity?, how meaningful is the monitoring river cross section with respect to the whole river reach?, do the measuring section belong to a straight patch of the river?, did the author check the presence of upstream boulders or pools which could bias the results?.

A: The description of methods has been expanded. Sections describing meteorological and hydrological measurements were added. Missing details of bedload transport measurements description were completed.

Figure 2. Based on which criteria the RBT modules were spaced along the cross section?

A: Proportional to the width of the trough at distances not exceeding 1.5-2 meters apart (line 182 in corrected version of the manuscript)

For how long the RBT modules were placed in the river to measure bedload fluxes?

A: ‘a continuous measurements at daily intervals (24 hours)’ (lines 176-177 and 186)

Which is the minimum and maximum sediment size collected by the RBT modules?.

A: ‘a maximum sample mass of up to 45-50 kg (optimum sample volume - up to 40% of container capacity - is about 18-20 kg).’ (lines 178-180)

Why 4 or 5 RBT modules (see Table 1) were chosen?

A: in 2011 the width of the channel in XS II was larger and in order to keep the distance between modules smaller than 2 m I added the 5th RBT module

- Provide the correct format to formula [1].

A: the format of the formula has been corrected

- Figure 4. Remove the 3D effect from the plot.

- Figure 4. Include a x-axis with the months. The current measurement-day x-axis is hard to interpret.

- Figure 4 needs to be interpreted in terms of annual precipitation and annual snow cover. This is the only way to distinguish any anomaly driven by the climatic forces.

A:  All comments considered. Unfortunately lack of hydrological data for 2011 resulted in lack of flow bars in this plot

- Results. Nothing is said about the discharge and sediment sizes. Is there any information on this matter?

A: Meteorological and hydrological background information has been added (except 2011 measurement period). Table 2 with the daily bedload fluxes has been added.

- Discussion. I think the author should explore the bedload hysteresis patterns associated to the unsteady flows [2,3]. Such way, each climate-derived mechanism of fluvial transport could be related to a specific type of sediment hysteresis.

A: The discussion was completed and these aspects were also discussed

Reviewer 4 Report

Sorry for the delay, was busy in the field. I have got the chance to go through the present version of the manuscript, which I find good on the whole. However, in my opinion the methodology and related presentation of data need an improvement, as getting to conclusions passes through those fundamental sections. In particular, some uncertainty analysis will further improve the range of applicability of the proposed methodology for bedload measurements. Interestingly, I see that the order of magnitude of the bedload fluxes is quite similar to some dense pyroclastic flows that resemble floods or dense water flows, which might be worthy of mention (Martí J, Doronzo DM, Pedrazzi D, Colombo F (2019) Topographical controls on small-volume pyroclastic flows. Sedimentology 66(6):2297–2317).

Best regards

Author Response

Response to the suggestions and comments of the Reviewer #4

Thank you for your expressed opinions and valuable comments. I tried to include them in the revised version as much as possible.

General changes to the text

  1. References were verified and supplemented
  2. All figures was reedited according to Reviewers’ comment
  3. The Table 2 has been added
  4. The text has been redrafted and supplemented where possible. The meteorological and hydrological description has been added.
  5. English language content was also corrected.

Round 2

Reviewer 1 Report

I thank the author for taking all my comments and suggestions into account and for including new data. The manuscript has certainly been improved. I also appreciate the detailed responses from the author. However, I’m still not fully convinced regarding the quality and novelty of the manuscript. I’m therefore sticking to my opinion from my last review.  

Reviewer 3 Report

The authors addressed all my previous queries.